# Wind Tunnel Testing of Plasma Actuator with Two Mesh Electrodes to Boundary Layer Control at High Angle of Attack [note 1]

**DOI:** 10.3390/s21020363

**Published:** 2021-01-07

**Authors:** Ernest Gnapowski, Jarosław Pytka, Jerzy Józwik, Jan Laskowski, Joanna Michałowska

**Affiliations:** 1Faculty of Technical Sciences, University College of Enterprise and Administration, 20-150 Lublin, Poland; 2Faculty of Mechanical Engineering, Lublin University of Technology, 20-618 Lublin, Poland; j.pytka@pollub.pl (J.P.); j.jozwik@pollub.pl (J.J.); 3Faculty of Management, Lublin University of Technology, 20-618 Lublin, Poland; jlasko@wp.pl; 4The Institute of Technical Sciences and Aviation, The State School of Higher Education, 22-100 Chełm, Poland; jmichalowska@pwsz.chelm.pl

**Keywords:** plasma actuator, flow control, wing airfoil, wind tunnel, mesh electrodes, DBD

## Abstract

The manuscript presents experimental research carried out on the wing model with the SD 7003 profile. A plasma actuator with DBD (Dielectric Barrier Discharge) discharges was placed on the wing surface to control boundary layer. The experimental tests were carried out in the AeroLab wind tunnel where the forces acting on the wing during the tests were measured. The conducted experimental research concerns the analysis of the phenomena that take place on the surface of the wing with the DBD plasma actuator turned off and on. The plasma actuator used during the experimental tests has a different structure compared to the classic plasma actuator. The commonly tested plasma actuator uses solid/impermeable electrodes, while in the research, the plasma actuator uses a new type of electrodes, two mesh electrodes separated by an impermeable Kapton dielectric. The experimental research was carried out for the angle of attack α = 15° and several air velocities *V* = 5–15 m/s with a step of 5 m/s for the Reynolds number Re = 87,500–262,500. The critical angle of attack at which the SD 7003 profile has the maximum lift coefficient is about 11°; during the experimental research, the angle was 15°. Despite the high angle of attack, it was possible to increase the lift coefficient. The use of a plasma actuator with two mesh electrodes allowed to increase the lift by 5%, even at a high angle of attack. During experimental research used high voltage power supply for powering the DBD plasma actuator in the voltage range from 7.5 to 15 kV.

## 1. Introduction

Guaranteeing a high level of aircraft safety requires many experimental tests of the aircraft’s components, including: landing gear [1,2,3,4], wings [5] and other components [6]. The manuscript presents and describes selected designs of DBD (Dielectric Barrier Discharge) plasma actuators, allowing us to increase flight safety by increasing the lift force. Plasma actuators DBD allow us to control the air flow in the boundary layer of the wing. It is a new solution for boundary layer control [7]. The DBD system does not have any moving parts such as rods, valves, diaphragms, cylinders or gears to activate it, thus creating no design problems that complicate the aircraft structure. One of the main mechanisms influencing the flow exerted by the DBD plasma actuator is the generation of “ion wind” [8,9]. As a consequence of flexible collisions between migrating charged particles and inert gas particles, they increase their energy, causing an “electric wind” that occurs in close proximity to the profile surface [10,11]. The construction of a plasma actuator is quite simple. It consists of two flat electrodes (the most commonly used metal/copper strips) that are separated by a dielectric. The dielectric prevents spark discharge that causes local increase in temperature. The first electrode is completely covered with a dielectric; the second electrode is directly exposed to the flowing air. Research by other scientists mainly focuses on changing the shape of solid copper electrodes into sawtooth, oval, rectangular and wave [12,13,14]. Changing the shape of the exposed electrode is aimed at increasing the discharge area [15,16]. These changes create a stronger “ion wind” affecting the air circulation in the boundary layer. The most frequently used frequency during experimental research is from several hundred Hz to several kHz [17,18,19,20]. The voltage range used during the experimental tests depends on the dielectric [21,22] used and ranges from a few kV to several dozen kV. The manuscript presents experimental studies of a plasma actuator with a new type of electrodes. Two mesh electrodes were used in the presented studies. The use of mesh electrodes in a plasma actuator is a new design solution. The presented tunnel tests of a plasma actuator with two mesh electrodes are a new solution, not described in the literature by other researchers. The most commonly used in experimental research are solid copper electrodes impermeable to air, [23,24,25,26,27,28,29]. In the presented experimental studies, the grounded electrode fully covered with a dielectric is much larger, it covers about 70% of the upper surface of the wing. The main purpose of the wind tunnel tests carried out to confirm the effectiveness of the new type of plasma actuator with two electrodes for controlling the boundary layer. The research was carried out for the wing model with the SD 7003 profile (made personally by the author of the manuscript) for an angle of attack of 15°; for such a high angle of attack, the profile loses lift under normal conditions. The critical angle of attack at which the SD 7003 profile has the maximum lift coefficient is about 11°.

## 2. Materials and Methods

Experimental research was carried out with the wing model with the SD 7003 profile, 250 mm wide and 250 mm long. The wing model is made of fiberglass, balsa and plywood. Two mesh electrodes are placed on the upper surface of the wing as shown in Figure 1. A large, grounded mesh electrode is 200 mm long and 200 mm wide is full covered with four layers of Kapton dielectric and is attached directly to the wing surface. The use of a dielectric to prevent spark discharges and high local temperature increase [30,31].

The use of mesh electrodes for the construction of a plasma actuator resulted from the previous experimental results of the author of the manuscript. Previous experimental research with plasma reactors with DBD discharges have shown that the use of mesh electrodes allows obtaining homogeneous discharges. The research carried out with DBD systems with small dimension mesh electrodes allowed for obtaining the highest efficiency of ozone generation during experimental research with plasma reactors. For the construction of plasma actuators, systems with the highest efficiency must be used, because the use of systems with low efficiency will increase the weight of the aircraft without the expected impact on the boundary layer.

The HV electrode is 200 mm long and 10 mm wide and is placed on the dielectric surface at a distance of 15 mm from the leading edge. Both mesh electrodes are made of AISI 304 stainless steel, it prevents oxidation of the electrode surface by ozone generated during DBD, the mesh size of the electrodes was 0.05 mm × 0.05 mm. The electrodes are connected to a high voltage power system with a value of several kV. The wing model was placed in the AeroLab wind tunnel (Aerolab LLC, Jessup, USA), which allows to control the air speed in the range of *V* = 4.5–65 m/s and measure the forces acting on the wing during experimental tests. The measuring stand and the wind tunnel are shown in Figure 2.

A typical DBD plasma actuator system with two solid electrodes is shown in Figure 3a. Figure 3b shows the setup and operation of the test system with two mesh electrodes.

Comparing the design of the plasma actuators in Figure 3a with classic plasma actuator with solid electrodes and Figure 3b with two mesh electrodes, differences in the structure and surface of the discharges can be seen. The use of mesh electrodes allows us to obtain a larger surface of discharges compared to the classic plasma actuator with solid electrodes; it is possible due to the use of an HV mesh electrode. Discharges in a plasma actuator with solid electrodes are generated at only one edge of the HV electrode. The use of mesh electrodes in the model presented in the manuscript allows for a larger area of discharges because the discharges occur at the two edges of the mesh electrode as well as directly through the surface of the HV electrode, which additionally increases the area of discharges. The entire surface of the high-voltage electrode forms one wide discharge.

Figure 4 compares the real design of the plasma actuator with solid electrodes in an asymmetric configuration Figure 4a and the tested plasma actuator with two mesh electrodes Figure 4b.

The experimental research presented in the manuscript were carried out for three air flow velocities of *V* = 5, 10 and 15 m/s and an angle of attack of 15°. During the experimental research, the air flow velocities were changed in steps of 5 m/s.

An autotransformer and a high voltage transformer 230/10,000 V 50 Hz were used to power the DBD plasma actuator located on the wing surface. Lissajous figures and high voltage were recorded during the experimental tests.

The measurements were performed with the Keysight DSO X 2012A 200 MHz, 2 GS/s oscilloscope (Keysight, Santa Rosa, CA, USA), equipped with a Tektronix P6015A high-voltage probe (Tektronix, Inc, Beaverton, OR, USA) and a Tektronix P2220 1:1/10:1 current probe (Tektronix, Inc, Beaverton, OR, USA). Figure 5 shows a diagram of the measurement system with individual elements marked. The internal structure of the wing model is also shown in Figure 5, and in particular the arrangement of the individual layers of the dielectric and electrodes. A photo of the real wing model used in the experimental tests is shown in Figure 1, with a description of the individual elements. The photo of the wing model shows a large, grounded electrode GND full covered with a dielectric; on the surface of the dielectric there is a high-voltage HV electrode.

During the experimental tests in the wind tunnel, photos of the wing model for individual air flow velocities were recorded using the Phantom V2511 (Vision Research, Wayne, NJ, USA) high-speed camera.

During the experimental research, the wing model was placed inside the AeroLab wind tunnel. The DBD plasma actuator is located on the upper surface of the wing model as shown in Figure 6. The wing model with the DBD plasma actuator was placed directly on the balancing force, which allows the registration of the forces acting on the model during experimental research. Figure 6 shows the wing model placed on the balancing force during experimental tests in the wind tunnel.

During the experimental research, a smoke generator was used, which allowed the observation of changes in air flow on the wing surface and the formation of turbulence.

## 3. Results

During the experimental research, use of the oscilloscope allowed for the recording of Lissajous figures with which the power of the discharges was calculated. Figure 7 shows an exemplary Lissajous figure for the discharge power of 1.0 W. The power of the discharges during the experimental research ranged of *P* = 1.0–2.2 W, with the supply voltage of *V_p_* = 7.5–15 kV. High voltage power supply with a frequency of 50 Hz was used to power the DBD plasma actuator.

During the tunnel tests, the forces acting on the wing model were also recorded, as shown in Table 1.

The Table 1 compares the forces acting on the wing model under the same conditions with the plasma actuator turn on and turn off. The uncertainty of measurements, calculated on the basis of the results of experimental research, is presented in Table 2.

In addition to the results presented in Table 1, the effectiveness of the plasma actuator is also confirmed by photos taken with a high-speed camera. The use of a smoke generator and high-speed camera allows to observe the air flow for the system with the plasma actuator turn on and turn off. The photos show the influence of the DBD system on the change of air flow over the upper surface of the wing. Turning on the plasma actuator allows for a more laminar airflow as shown in Figure 8b. During the experiment, the camera recorded photos at 3000 frames per second, which provided high quality photos.

The presented results of experimental tests in the wind tunnel on the SD 7003 wing model show the effectiveness of the DBD plasma actuator with two mesh electrodes, which is confirmed by the results presented in Table 1 and the tunnel photos in Figure 8, Figure 9 and Figure 10. Experimental research were carried out for a high angle of attack for which the SD 7003 profile wing loses its lift under normal conditions (critical angle of attack about 11°).

Figure 8a shows the wing model with the plasma actuator turned off, with visible non-laminar airflow. The point of separation and turbulence are especially visible at the end of the wing near the trailing edge. Turning on the plasma actuator at the air velocity of *V* = 5 m/s in Figure 8b allowed to significantly reduce the turbulence that is visible in Figure 8a. The airflow in Figure 8b is not completely laminar but much less turbulent than the flow with the plasma actuator off Figure 8a. The tunnel photo shown in Figure 9a for an air velocity of 10 m/s and Figure 10a for an air velocity of 15 m/s for the configuration with plasma actuator turned off is non-laminar with visible turbulence, while in Figure 9b and Figure 10b with plasma actuator turned on, the airflow is more laminar at air flow velocity of *V* = 10 m/s and *V* = 15 m/s. Comparing Figure 8b, Figure 9b and Figure 10b (Plasma On), we can see the negative effect of airflow velocity on the efficiency of the plasma actuator. In Figure 9b and Figure 10b the air jets do not adhere to the wing, which results in a reduced increase in lift.

The lift coefficient C_L_ was used to determine the lift force changes depending on the air flow velocity and the angle of attack. Figure 11 shows the change in the lift coefficient C_L_ of the wing model with the plasma actuator turn off and turn on. The changes in the lift coefficient presented in Table 3 are most visible on the graph for the flow velocity in the range of *V* = 5–10 m/s, when the air flow velocity is low.

Figure 12 and Table 4 presents changes in resultant force depending on air flow velocity in the range of 5 to 15 m/s and angle of attack 15°.

The results of the experimental tests presented in Table 4 show the change of the resultant force trend for the air flow velocity in the range from *V* = 5 m/s to *V* = 15 m/s for the angle of attack of 15°.

The changes in the resultant force presented in the diagram in Figure 12 are small because the changes in the lift coefficient of the tested SD 7003 wing profile are small because the critical angle of attack for this profile is 11°. In the experimental tests, the experiment was carried out for an angle of 15° greater than the critical angle of attack. At an angle of attack greater than 11°, the SD 7003 profile loses total lift and stall occurs.

The use of a plasma actuator allowed to obtain a positive increase in the lifting force as well as the resultant force even in such unfavorable conditions.

The main factor influencing the increase in the lift force and the efficiency of the plasma actuator with two mesh electrodes is the increase in the discharge area. The discharge on the high-voltage mesh electrode is created over the entire surface of the electrode, creating a wide discharge belt (in a classic plasma actuator with a solid electrode, the discharge is generated only on one edge). The design solution presented in the manuscript is a new solution for changing the geometry of the electrodes and is not used by other researchers.

The experimental studies show that the efficiency of the plasma actuator depends on several factors, including air flow velocity *V*, as well as the parameters of the plasma actuator supply system and the power of the discharges. Many experimental studies confirm that the efficiency of DBD systems decreases with the increase in gas flow velocity (e.g., ozone generators, the concentration of ozone decreases) [32,33,34], the same is the case with DBD plasma actuators [35,36] because the operation of the plasma actuator is precisely to generate the plasma “ion wind”.

## 4. Conclusions

The results of the conducted experimental research confirm the effectiveness of the new type of plasma actuator with two mesh electrodes. Tunnel tests of the wing model with the SD7003 profile allowed to obtain an increase in the lift coefficient by 5% at a high angle of attack of 15° (critical angle of attack 11°), which is confirmed by Figure 11 and Table 3. The high efficiency of the tested design of the plasma actuator is possible because the discharges appear at both edges and directly through the surface of the high voltage electrode. These phenomena do not occur in the classic solution with impermeable solid electrodes, in which the discharges are only on one edge, discharges directly through the surface of the HV solid electrode are impossible. The geometry of the electrodes is a key factor in the efficiency of the plasma actuator. Increasing the discharge area allows for a higher gas ionization density, affecting the increase in the lift coefficient C_L_. Many researchers change the shape of the solid electrodes to increase the efficiency of the plasma actuator. The mesh electrode increases the effective surface of the discharges, thus increasing the efficiency of the plasma actuator with two mesh electrodes. Another important factor is the supply system, especially the supply voltage and frequency. Both of these factors directly affect the power of the discharges. The plasma actuator with two mesh electrodes described in the manuscript is a new design that requires further optimization.

The conducted experimental research and the obtained results show the potential of the new design of the plasma actuator. In order to determine the potential of the new design of the plasma actuator, it requires further experimental research to determine the optimal configuration of the system, one of the factors that should be examined is the determination of the mounting location and the number of HV electrodes.

## Figures and Tables

**Figure 1 sensors-21-00363-f001:**
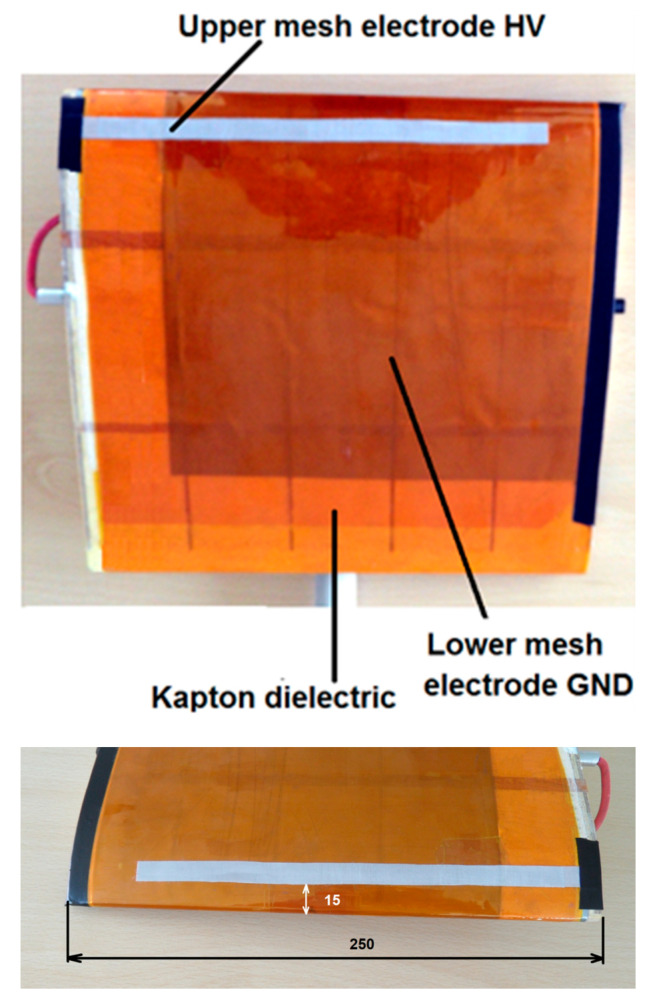
Wing model with the profile SD7003, with a visible plasma actuator with two mesh electrodes.

**Figure 2 sensors-21-00363-f002:**
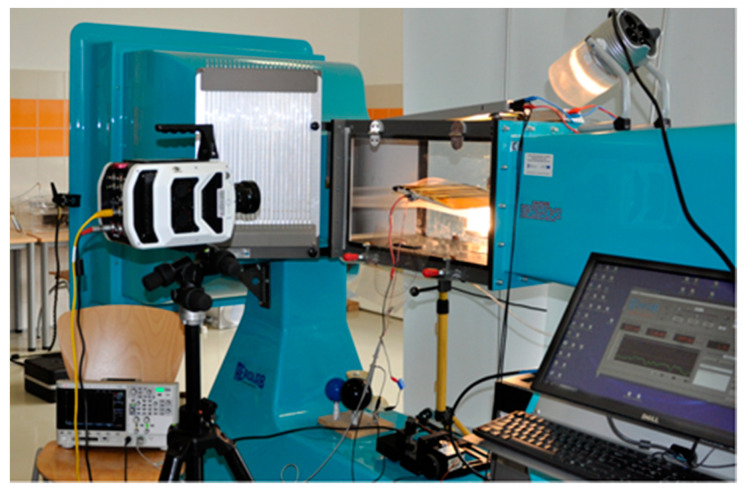
Measurement stand with equipment.

**Figure 3 sensors-21-00363-f003:**
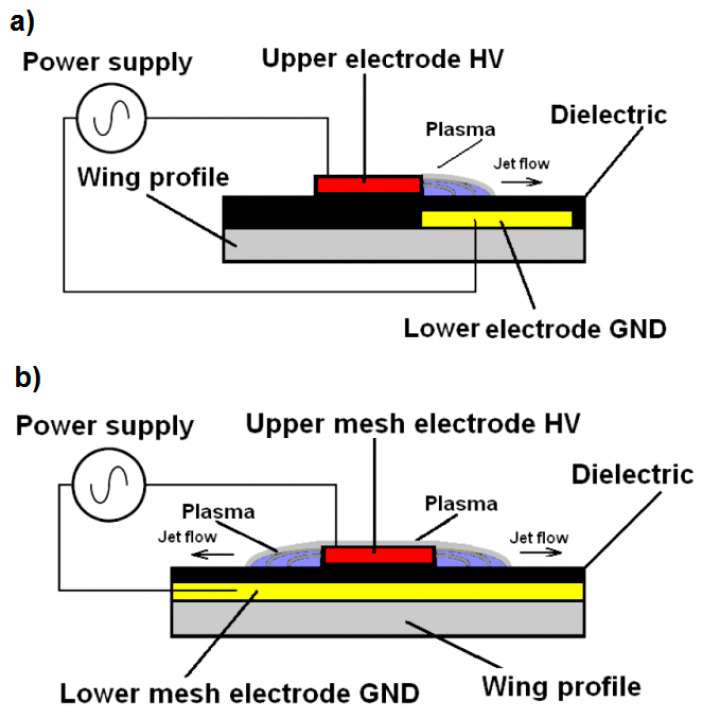
Construction of plasma actuators: (**a**) classic plasma actuator with solid electrodes with visible discharges only on one edge, (**b**) plasma actuator with two mesh electrodes with discharges visible on two edges and directly through the surface of the HV electrode.

**Figure 4 sensors-21-00363-f004:**
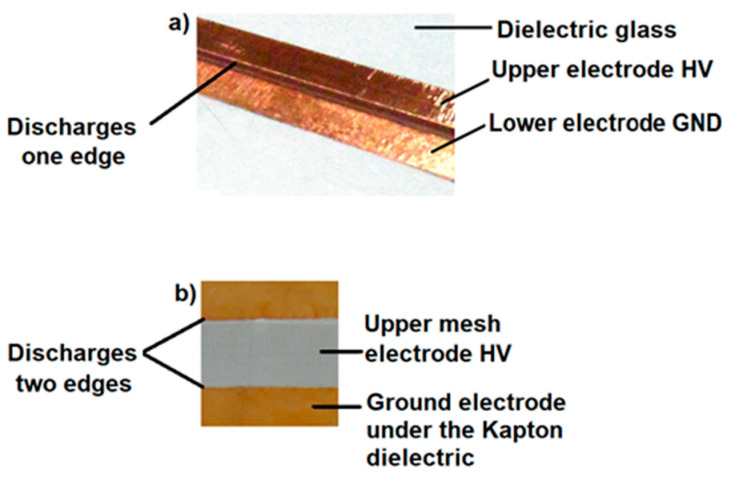
Comparison of the design of plasma actuators: (**a**) classic with solid electrodes and, (**b**) with two mesh electrodes.

**Figure 5 sensors-21-00363-f005:**
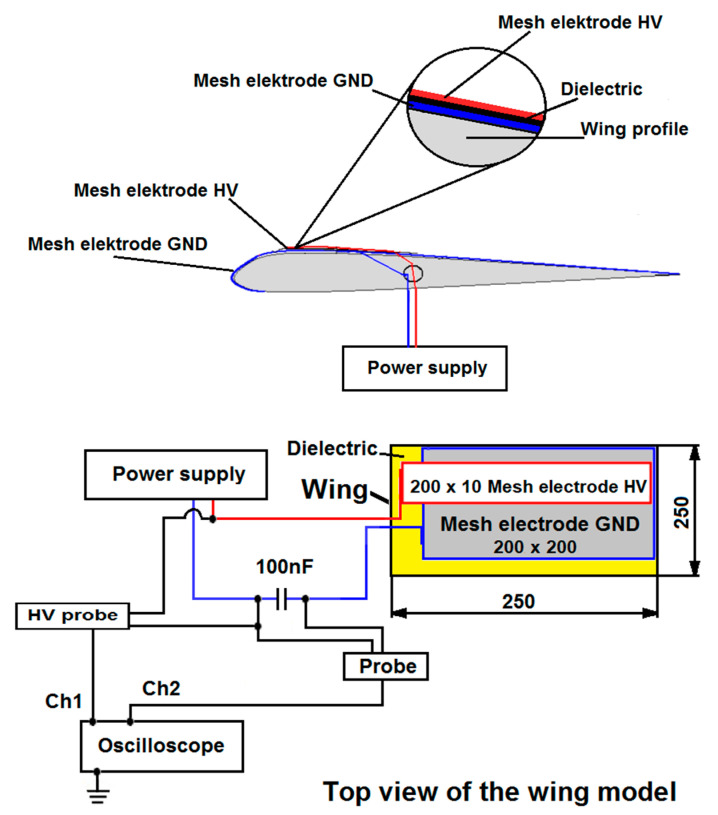
Measurement diagram and electrode system arranged on the wing model.

**Figure 6 sensors-21-00363-f006:**
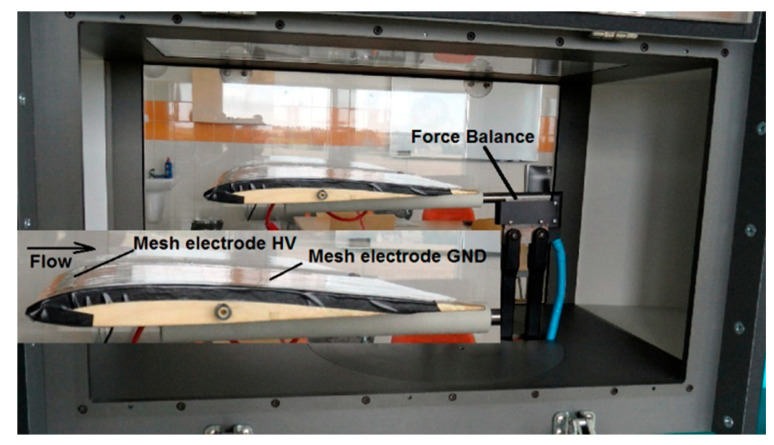
Wing model placed inside the wind tunnel.

**Figure 7 sensors-21-00363-f007:**
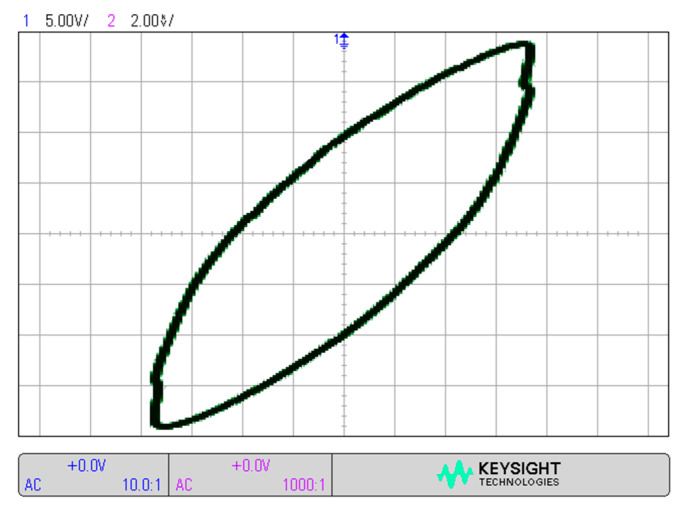
Lissajous figure for *P* = 1.0 W discharge power, at *V_p_* = 7.5 kV supply voltage.

**Figure 8 sensors-21-00363-f008:**
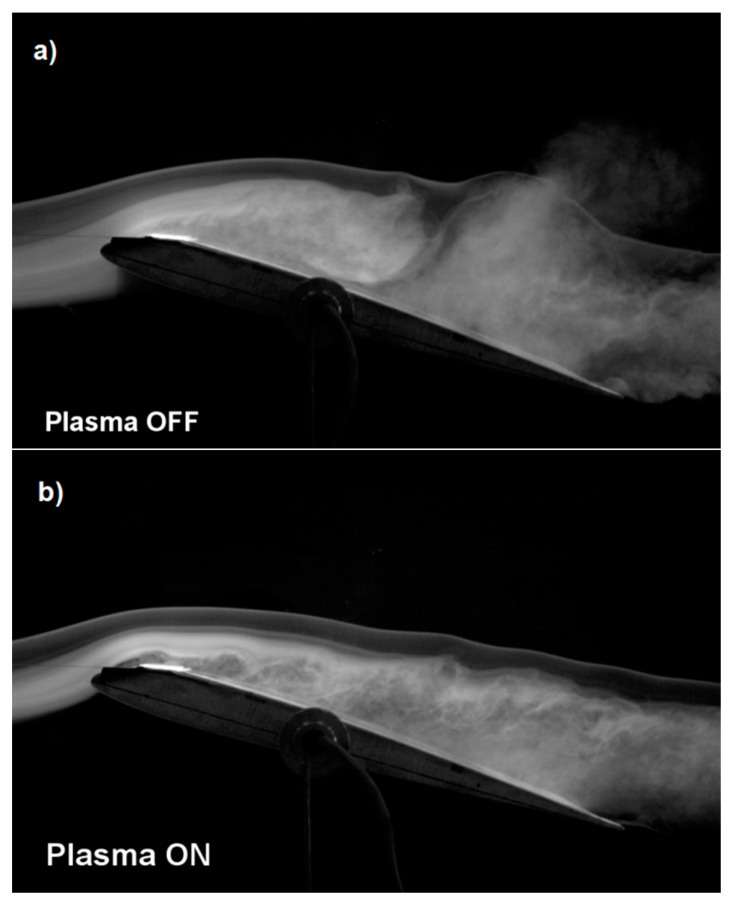
Tunnel photos of the wing model with the SD 7003 profile for the angle of attack α = 15° at the air flow velocity *V* = 5 m/s for; (**a**) plasma actuator turned off and (**b**) plasma actuator turned on.

**Figure 9 sensors-21-00363-f009:**
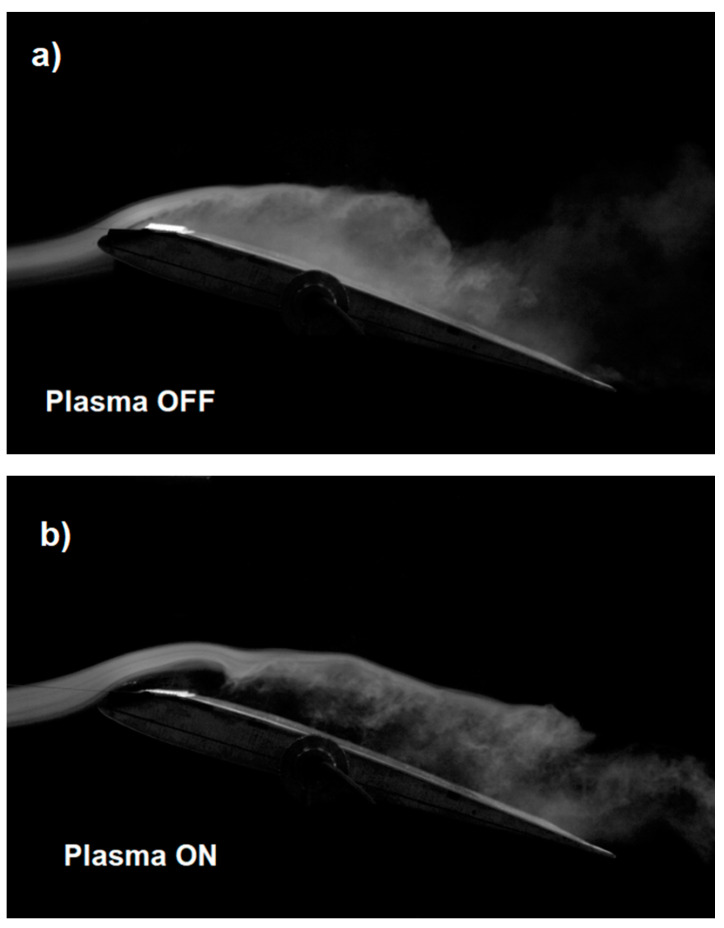
Tunnel photos of the wing model with the SD 7003 profile for the angle of attack α = 15° at the air flow velocity *V* = 10 m/s for; (**a**) plasma actuator turned off and (**b**) plasma actuator turned on.

**Figure 10 sensors-21-00363-f010:**
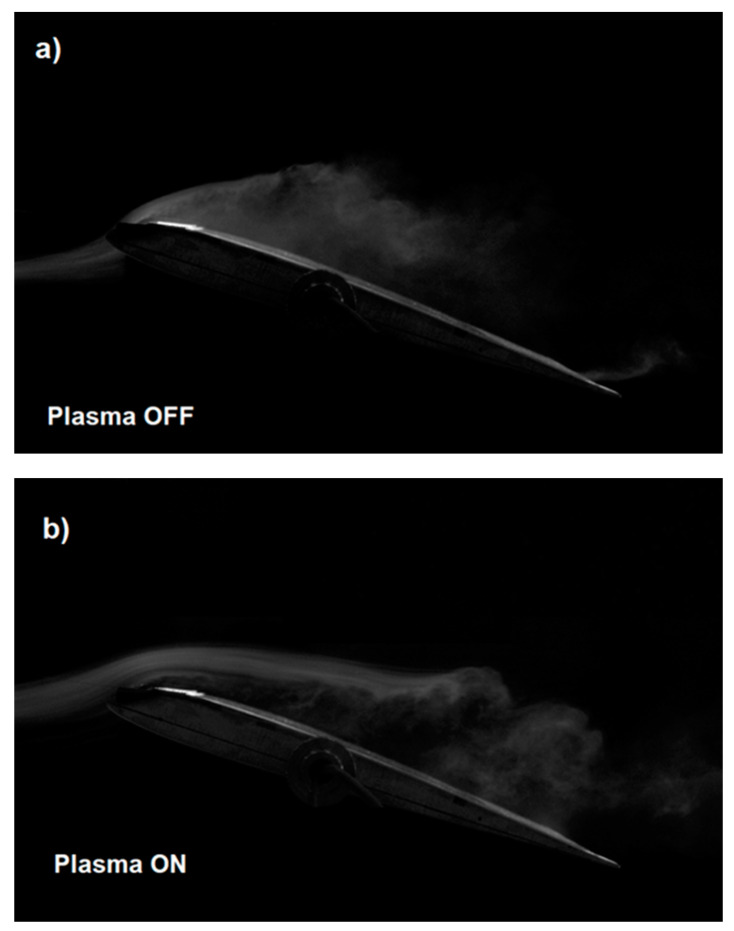
Tunnel photos of the wing model with the SD 7003 profile for the angle of attack α = 15° at the air flow velocity *V* = 15 m/s for; (**a**) plasma actuator turned off and (**b**) plasma actuator turned on.

**Figure 11 sensors-21-00363-f011:**
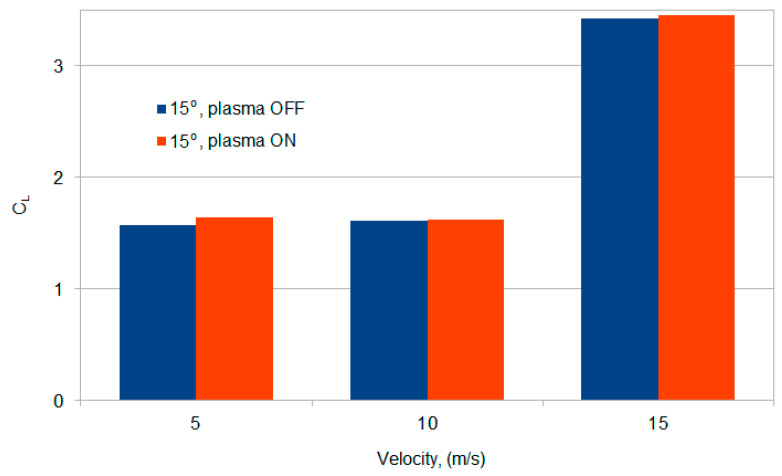
Graph of changes in the trend of the lift force coefficient for the wing model for the angle of attack α = 15° and the air flow speed in the range of *V* = 5–15 m/s, when the plasma actuator turn on and turn off.

**Figure 12 sensors-21-00363-f012:**
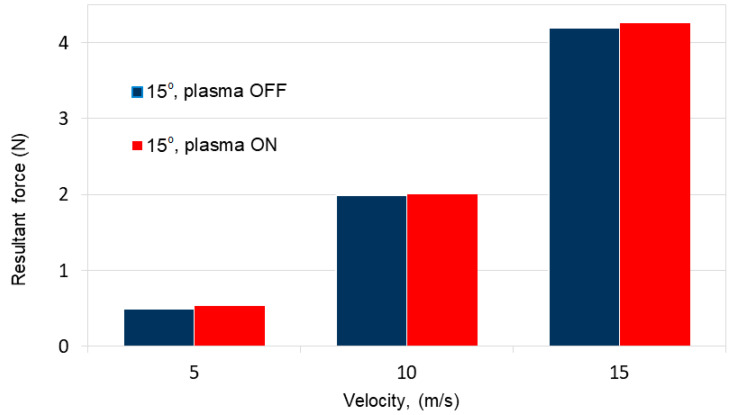
Changes in the resultant force trend depending on air flow velocity in the range of 5–15 m/s and angle of attack 15°.

**Table 1 sensors-21-00363-t001:** Normal and axial force measurements at angle of attack of 15°.

*V* (m/s)	Normal Force (N)		Axial Force (N)	
	Plasma actuator			
	OFF	ON	OFF	ON
5	1.52	1.57	0.08	0.09
10	6.46	6.46	0.31	0.31
15	13.35	13.48	0.65	0.66

**Table 2 sensors-21-00363-t002:** The uncertainty measured of forces for angle of attack of 15°.

	Measurement Uncertainty			
	Normal Force (N)		Axial Force (N)	
	Plasma actuator			
*V* (m/s)	OFF	ON	OFF	ON
5	0.007	0.013	0.002	0.006
10	0.003	0.007	0.002	0.006
15	0.083	0.075	0.004	0.008

**Table 3 sensors-21-00363-t003:** Change of the lift coefficient C_L_ for the angle of attack of 15°.

*V* (m/s)	Lift Coefficient C_L_	
	Plasma actuator OFF	Plasma actuator ON
5	1.57	1.62
10	1.61	1.62
15	3.42	3.44

**Table 4 sensors-21-00363-t004:** Changes in resultant force depending on air flow velocity and angle of attack 15°.

*V* (m/s)	Resultant Force (N)	
	Plasma actuator OFF	Plasma actuator ON
5	0.5	0.53
10	1.98	2
15	4.19	4.25

## Data Availability

Data is contained within the article.

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
