# Peer review of "Wind Tunnel Testing of Plasma Actuator with Two Mesh Electrodes to Boundary Layer Control at High Angle of Attack†"

_sensors, 2021, doi:10.3390/s21020363_

Round 1

Reviewer 1 Report

Dear authors,

The manuscript sensors-1031817-peer-review-v1, entitled ‘Wind Tunnel Testing of Plasma Actuator with Two Mesh Electrodes to Boundary Layer Control at High Angle of Attack’ presents experimental studies of a plasma actuator with a new type of electrodes (mesh). The presented tunnel tests of a plasma actuator with two mesh electrodes are a new solution (originality of the work).

The methodology is well presented, as well as the techniques used (electrical characterization of plasma actuator, tunnel photos, or lift coefficient determination). The experimental findings are well commented and supported by good quality figures. The plasma source, an asymmetric DBD (HV electrode 200x10 mm, and GR electrode 200x200 mm) with mesh electrodes, was ignited using a high voltage between 7-15 kV, resulting in a ~ 3 W power. Proper diagnosis equipment was used for the electric and high-speed photography.

The conclusions are in brief but sustained by the experimental findings. In their experimental studies, the authors used the grounded electrode fully covered with a dielectric (with a much larger covered area, of about 70% of the upper surface of the wing). The research was carried out for the wing model with the SD 7003 profile for an angle of attack of 15 degrees, for such a high angle of attack, the profile loses lift under normal conditions. The critical angle of attack at which the SD 7003 profile has the maximum lift coefficient is about 11 degrees. By these, an increase in the lift coefficient by 5% at a high angle of attack of 15 degrees was obtained by the authors. Moreover, they shown that the geometry of the electrodes is a key factor in the efficiency of the plasma actuator. Furthermore, by increasing the discharge area allows for a higher gas ionization density, thus affecting the increase of the lift coefficient. In the end, the authors revealed that another important factor is the supply system, especially the supply voltage and frequency. Both factors directly affect the electrical power of the discharges. This plasma actuator, with two mesh electrodes described in the manuscript, is a new design that requires further optimization.

I hereby propose this manuscript to be considered for publication AS IS in SENSORS journal.

Author Response

Thank you very much for reviewing our manuscript. We are also glad that the manuscript meets all expectations related to its publication and that it does not require further corrections.

Yours sincerely, the authors of the manuscript.

Reviewer 2 Report

The paper describes experimental tests on a wing profile in which the flow is controlled by means of DBD actuators. I suggest the following modifications to improve the work:

  • In Section 2 the authors explain that they chose to use mesh electrodes instead of classical electrodes. They should provide in this section the motivation that lead them to this choice.
  • The Figure 3 shows two configurations of plasma actuators: the classical one with one jet and the symmetric one with two opposite jets. The authors should explain why the symmetric configuration should be considered: the presence of two jets in opposite directions does not tend to reduce the net effect?
  • Line 115: the sentence should be re-written as "The DBD plasma actuator is locate on the upper surface of the wing model as shown in Figure 6"
  • The results in Table 1 should be provided together with the uncertainties on the measured quantities. The uncertainty on the measured force is critical in this application. Furthermore, the authors should provide information on the averaging window size used to provide the force values ( the photos show a clearly unsteady field so I suppose that the provided force values are time-averaged)
  • In Figure 8,9 and 10 the authors provide instantaneous visualisations of the flow field. It is quite difficult to get an objective conclusion from these results because the conclusion could be strongly influenced by the choice of the snapshot. The author could consider the possibility to acquire a large number of snapshots and then to average them in order to get an experimental visualisation of the average field. If this is not possible, the authors should comment on this point.
  • Lines 162 and 162 : the sentence "The laminar flow is shown in Figure 8b with the plasma actuator turn on ..." is not strongly supported by the results: a comparison between Figure 8a and 8b shows that both are unsteady flows with some noticeable vortex structures. The sentence "Turbulence is not even visible at the trailing edge" has the same problem: how is it possible to get this conclusion from Figure 8a and 8b?
  • Figure 11: the results show a smooth and continuous variation of the lift coefficient. Is this obtained with just 3 experimental points and performing a fitting? Maybe it would be better to increase the number of experimental points to get a more significant trend.
  • Figure 12: the authors should report in the text the uncertainty related to force measurements otherwise it is difficult to evaluate the effects of the DBD.s
  • Line 209: maybe instead of "plasma activator" the authors mean "plasma actuator"

Author Response

Thanks for your comments and suggestions on our manuscript, we have made corrections according to the comments. Let us include the answers to the question and the manuscript with the marked corrections.

1) In Section 2 the authors explain that they chose to use mesh electrodes instead of classical electrodes. They should provide in this section the motivation that lead them to this choice.

Thank you for your comment. The use of mesh electrodes for the construction of a plasma actuator resulted from the previous experimental results of the author of the manuscript. Previous experimental research with plasma reactors with DBD discharges have shown that the use of mesh electrodes allows obtaining homogeneous dischargess. The research carried out with DBD systems with small dimension mesh electrodes allowed for obtaining the highest efficiency of ozone generation during experimental research with plasma reactors. For the construction of plasma actuators, systems with the highest efficiency must be used, because the use of systems with low efficiency will increase the weight of the aircraft without the expected impact on the boundary layer. We have made the necessary corrections to the manuscript text.

2)  The Figure 3 shows two configurations of plasma actuators: the classical one with one jet and the symmetric one with two opposite jets. The authors should explain why the symmetric configuration should be considered: the presence of two jets in opposite directions does not tend to reduce the net effect?

Thank you for your comment. Comparing the design of the plasma actuators in Figure 3a with the asymmetric system of electrodes and Figure 3b with two mesh electrodes, differences in the structure and surface of the discharges can be seen. The use of mesh electrodes allows to obtain a larger surface of discharges compared to the classic asymmetric system of electrodes, it is possible due to the use of an HV mesh electrode. Discharges in a plasma actuator with an asymmetric system are generated at only one edge of the HV electrode. The use of mesh electrodes in the model presented in the manuscript allows for a larger area of discharges because the discharges occur at the two edges of the mesh electrode as well as directly through the surface of the HV electrode, which additionally increases the area of discharges. The entire surface of the high-voltage electrode forms one wide discharge..

3) Line 115: the sentence should be re-written as "The DBD plasma actuator is locate on the upper surface of the wing model as shown in Figure 6"

Thank you for your comment, we have made a correction as recommended.

4) The results in Table 1 should be provided together with the uncertainties on the measured quantities. The uncertainty on the measured force is critical in this application.

We are grateful to the reviewer for pointing out this concern. We made the calculation of the measurement uncertainty based on our measurement results. The calculated uncertainty of measurements is presented in Table 2. Table 1 shows the values of the normal and axial force measured during the experimental research, these are average values.

5) Furthermore, the authors should provide information on the averaging window size used to provide the force values ( the photos show a clearly unsteady field so I suppose that the provided force values are time-averaged).

In experimental studies, the measurement of forces and the recording were made for the instantaneous values that appeared during the experiment. They were not carried out at specified intervals but at any time intervals. The time intervals between the individual measurements were introduced randomly to avoid random repeatability of the results of the measured forces.

6) In Figure 8,9 and 10 the authors provide instantaneous visualisations of the flow field. It is quite difficult to get an objective conclusion from these results because the conclusion could be strongly influenced by the choice of the snapshot. The author could consider the possibility to acquire a large number of snapshots and then to average them in order to get an experimental visualisation of the average field. If this is not possible, the authors should comment on this point.

Thank you for your comment. We’ve observed very similar images of the airstream during the wind tunnel tests and have chosen the snapshots included in the manuscript. They are representative and show a significant trend for the particular test variation. Of course, we’ve acquired lots of snapshots, but averaging them would make the illustration unclear. Typically, wind tunnel tests visualization is performed with the use of separate snapshots, as we’ve done it in our paper.

7) Lines 162 and 162 : the sentence "The laminar flow is shown in Figure 8b with the plasma actuator turn on ..." is not strongly supported by the results: a comparison between Figure 8a and 8b shows that both are unsteady flows with some noticeable vortex structures. The sentence "Turbulence is not even visible at the trailing edge" has the same problem: how is it possible to get this conclusion from Figure 8a and 8b?

Thank you for your comment. The comparison presented in Figures 8a and 8b was to show the increase in air flow laminarity with the plasma activator turned on and the system with the plasma activator turned off. We apologize for the inaccurate description of our intentions. We have made the necessary corrections to the manuscript text.

8) Figure 11: the results show a smooth and continuous variation of the lift coefficient. Is this obtained with just 3 experimental points and performing a fitting? Maybe it would be better to increase the number of experimental points to get a more significant trend.

Thank you for the comment, which is trurly right. We’ve changed the type of graph in Fig. 11 in order to show a trend and not try to derive a relationship and curve fitting. Three points is not enough for a relationship but one can observe and evaluate a trend based on these results, the more that the trend is very clear. Moreover, we’re respecting your comment for our future reseach plans – we’re planning to conduct our wind tunnel tests with an increased number of angle of attack input settings.

9) Figure 12: the authors should report in the text the uncertainty related to force measurements otherwise it is difficult to evaluate the effects of the DBD.s

Thank you for the comment. We added information about the uncertainty of force measurement in the manuscript, according to the reviewer's comment.

10) Line 209: maybe instead of "plasma activator" the authors mean "plasma actuator"

Thanks for the comment, we apologize for the inconvenience, we have made a correction.

Thank you for your comments and suggestions regarding our manuscript. Sincerely, authors of the manuscript.

Reviewer 3 Report

I reviewed the paper entitled “Wind Tunnel Testing of Plasma Actuator with Two Mesh Electrodes to Boundary Layer Control at High Angle of Attack” with an interest.  

In this paper the authors reported the results of experimental study of  the wing model with the SD 7003 profile. A plasma actuator with DBD discharges was placed on the wing surface to control boundary layer. The experimental tests were carried out in the AeroLab wind tunnel where the forces acting on the wing during the tests were measured. The conducted experimental research concerns the analysis of the phenomena that take place on the surface of the wing with the DBD plasma actuator turned off and on. The plasma actuator used during the experimental tests has a different structure compared to the classic plasma actuator.

In general, this manuscript is interesting. The results are useful for researchers and engineers. Overall, the paper is well-structured. I would recommend this manuscript for publication in Sensors after minor revision according to following recommendations:

  • In the first place, I would encourage the authors to extend the abstract more with the key results. As it is, the abstract is a little thin and does not quite convey the interesting results that follow in the main paper.
  • "Discussion" section should be edited in a more highlighting, argumentative way. The author should analysis the reason why the tested results is achieved.
  • Conclusion section should be rearranged. According to the topic of the paper, the authors may propose some interesting problem as future work in conclusion.

Author Response

Thanks for your comments and suggestions on our manuscript, we have made corrections according to the comments. We attach the answers to the question and the manuscript with the marked corrections.

1) In the first place, I would encourage the authors to extend the abstract more with the key results. As it is, the abstract is a little thin and does not quite convey the interesting results that follow in the main paper

Thank you for comment. We have revised the manuscript according to sugestaimi. In the abstract, we have introduced additional descriptions with included key results. The introduced corrections are marked in the text of the manuscript.

2) "Discussion" section should be edited in a more highlighting, argumentative way. The author should analysis the reason why the tested results is achieved.

Thank you for your comments, we have made corrections to the manuscript in line with the reviewer's suggestion. We highlighted the reason why our plasma actuator has increased performance compared to the classic solid electrode system. The main factor that allows to increase the lift force and the efficiency of the plasma actuator with two mesh electrodes is the increase of the discharge area. The discharge on the high-voltage mesh electrode is created over the entire surface of the electrode, creating a wide discharge belt (in the classic arrangement with a fixed electrode, the discharge is only one narrow edge). The discharges occur at both edges of the HV electrode and directly through the surface of the mesh electrode. The use of mesh electrodes, especially the high-voltage mesh electrode, allowed to increase the lift coefficient CL. Increasing the surface area of the discharges increases the amount of plasma produced that creates the ion wind.

3) Conclusion section should be rearranged. According to the topic of the paper, the authors may propose some interesting problem as future work in conclusion

We agree with the reviewer's suggestion for the necessary summary corrections, we made corrections according to the reviewer's suggestions. We have also added the suggested directions for further research on the optimization of a plasma actuator with two mesh electrodes. The main direction of future research should be to determine the number and position of HV electrodes on the wing surface. Further optimizations should concern, for example, changes to the power supply system. Corrections and changes to the text were also placed and marked in the text of the manuscript

Thank you for your comments and suggestions.

Sincerely, authors of the manuscript.

Round 2

Reviewer 2 Report

The authors significantly improved the paper and so I suggest the pubblication in the present form.